# Integrating postpartum IUD counselling and insertion into routine maternity care in Nepal: Assessing trends over time

**Mahesh Chandra Puri**[1]*, **Muqi Guo**[2], **Lucy Stone**[3], **Iqbal H. Shah**[2]

**1** Center for Research on Environment, Health and Population Activities (CREHPA), Kusunti, Lalitpur, Kathmandu, Nepal, **2** Harvard T. H. Chan School of Public Health, Boston, Massachusetts, United States of America, **3** Faculty of Medicine, Health and Life Science, Swansea University Medical School, Swansea University, Wales, United Kingdom

* mahesh@crehpa.org.np

**Data Availability Statement:** The dataset used and/or analyzed for this paper is publicly available from https://dataverse.harvard.edu/dataset.xhtml?persistentId=doi:10.7910/DVN/QK02R8.

## Abstract

To meet the postpartum family planning (PPFP) needs of women in Nepal, an intervention was launched to integrate PPFP counselling and postpartum IUD (PPIUD) insertion into maternity care. Women delivering in study hospitals over a period of 18 months were interviewed at the time of delivery and at 15 months following the end of the study enrollment period to assess if the impact of the intervention observed at the end of the study was maintained. Data were collected prior to the intervention, at the middle month of the intervention roll out, at the end of the enrollment period and 15 months after the end of the enrollment period. We compared PPFP counselling and insertion rates before, during, at the end of and after the intervention study period, using cross-tabulation and chi-square tests. Overall, PPFP counselling rates increased from 11% at the baseline month to 45% at the end of the enrollment in February 2017 and remained the same 15 months later in July 2018. PPIUD uptake, however, rose from a negligible 0.1% at the baseline to 4.3% in February 2017, but declined to 3.4% in July 2018. PPIUD uptake among women who were counselled showed a similar trend, increasing from 1.9% at the baseline to 9.6% in February 2017 and declining to 6.0% in July 2018. The intervention had an appreciable continued impact on PPIUD counselling rates and although PPIUD uptake rose during the intervention, this trend was not observed in the 15 months post-study follow up. The impact of the intervention was greater and persistent in hospitals that had a longer period of exposure to intervention. The results suggest that counselling was well integrated with the maternity care, though uptake of PPIUD dropped after intervention activities such as active monitoring, technical supervision, provision of IUDs and training were withdrawn.

**Trial registration:** This study has been registered with Clinical Trial.gov. The registration number is NCT 02718222. Details about the study design have been published by Canning et al, 2016.

**Funding:** Though the data collection of this study was funded by Susan Thompson Buffett Foundation (via grant 4041), the funder had no role in the design, implementation, data collection, analysis, and interpretation of results or in the dissemination of findings. The authors did not receive any financial support to write this paper.

**Competing interests:** The authors have declared that no competing interests exist.

## Introduction

The ability to decide family size and spacing between births is a basic human right [1]. The Programme of Action launched at the International Conference on Population and Development (ICPD) in 1994 and adopted by 179 United Nations (UN) member states subsequently promoted this right as a fundamental reason for increasing access to family planning (FP) globally [2,3]. Poor access to FP and the inability to space births effectively are intrinsically linked to increased maternal and child mortality and morbidity, especially in low- and middle-income countries (LMICs) [4–9]. Improvements in access to FP and modern methods of contraception in LMICs have been made in recent years, however, more needs to be done to ensure the right to desired birth spacing is realised, especially in light of concerns about the effects the 2019 Novel Coronavirus (COVID-19) could have on the progress made in the area of sexual and reproductive health and rights [10,11].

While contraceptive use in Nepal has improved in recent decades, there is a clear unmet need for postpartum family planning (PPFP) counselling and contraceptive use among postpartum women. In 2016, only 13% of women in Nepal received FP counselling during their postpartum period [12]. The World Health Organisation (WHO) recommends that women have a birth to pregnancy interval of at least 2 years to reduce the risk of negative maternal and child health outcomes [13]. Despite this, one in two women in Nepal have an unmet need for FP within 24 months postpartum and 21% of births occurred within the two-year postpartum period [12,14].

One intervention to address the gap in PPFP counselling and uptake of PPFP in Nepal and other LMICs was to integrate PPFP services into existing routine maternity care, particularly antenatal care (ANC) and institutional delivery [15,16]. Rates of ANC visits and institutional delivery in Nepal have improved in recent years. In 2016, 84% of women who had given birth had at least one ANC visit, up from 59% in 2011 and 57% of all births took place in a health facility, up from 35% in 2011 [12]. As many women in Nepal have difficulties accessing health facilities and trained health providers [17], improvements in the use of maternity services created an opportunity for the intervention to utilise the services women were already accessing to 1) provide PPFP counselling during ANC visits and 2) insertion of an immediate postpartum intrauterine device (PPIUD) after an institutional delivery to women interested in long-term contraceptive protection [15,16].

A PPIUD that is inserted within 48 hours after birth is a long-acting reversible contraceptive (LARC) method that has been proven to be reliable, safe and cost-effective, and can be made readily available to women delivering in health facilities, making it an attractive option for women in Nepal [18–21]. Despite the recognised benefits, in 2016 only 1.4% of married women of reproductive age (15–49 years) reported using an IUD and 28.2% of those using an IUD discontinued it within 12 months after insertion [12]. This low rate in IUD uptake has been attributed to several barriers facing many women in Nepal. Alongside limited provision of IUDs and limited health facilities and providers in more rural areas of Nepal, there is also a lack of staff trained in PPFP counselling and IUD provision, poor knowledge among women about the IUD, and pre-exiting biases towards the IUD by both providers and women and their families [17,22,23].

The PPIUD intervention in Nepal aimed to train providers on PPFP and PPIUD counselling and insertion techniques to improve providers' knowledge and skills with the wider aim of improving patient knowledge and contraceptive options through PPFP counselling and increasing the uptake of IUDs among postpartum women [15]. Previous studies analysing the impact of PPIUD services have been based on cross-sectional data. One study was an initiative that integrated PPFP and PPIUD services into maternity care in six countries (Ethiopia,

Guinea, India, Pakistan, Philippines, and Rwanda). The findings from this study suggested that integrating PPFP and PPIUD provision at the point of delivery improved access to contraception. Among women counselled, between 2.3% and 5.8% accepted a PPIUD and a median of approximately 2% of all women who had a delivery in the study facilities chose a PPIUD [24].

One previous study that assessed the PPIUD intervention in Nepal investigated the causal impact of PPFP counselling and PPIUD insertion in routine maternity services using a randomised stepped-wedge cluster design [15]. Using data for women who gave birth at the study hospitals over an 18-month period (between 8th September 2015 and 8th March 2017), the study assessed the trends in PPIUD counselling and insertion. The study showed a 25-percentage point increase in PPIUD counselling and a 4.4 percentage point increase in PPIUD uptake among women exposed to the intervention. The study also estimated that on average, being counselled as part of the intervention could increase the uptake of PPIUDs by approximately 17 percentage points [15]. Using the same data as the previous study, our study adds to the findings by investigating the impact the intervention had at different points during the study period and whether the intervention effects differed between the different groups of hospitals within the study. This study also aims to examine the longer term intervention effects using follow up data for the month of July 2018 (15 months after the end of the evaluation study enrolment when most of the intervention activities were no longer operational). This follow up data will give an indication of whether PPIUD counselling, insertion and insertion among those counselled has increased or decreased since the original analysis. It is important to investigate the longer-term impact of the intervention to examine if the outcomes observed at the end of the intensive intervention were maintained 15 months later when most of its active elements were no longer operational. Such understanding will assist with the design of programs and strategies to integrate PPFP and PPIUD counselling and insertion into existing antenatal and institutional delivery services in Nepal.

## Data and methods

### The intervention and evaluation

The PPIUD initiative by the International Federation of Gynecology and Obstetrics (FIGO) promoted the provision of immediate PPIUD services in Nepal and five other countries–India, Sri Lanka, Tanzania, Kenya, and Bangladesh [25]. The aim of FIGO PPIUD initiative was to address the gap in the continuum of maternal health care and to provide for the postpartum contraceptive needs of women by increasing the capacity of health care professionals to offer PPIUDs by training community midwives, health workers, doctors and delivery unit staff, as appropriate [25]. In Nepal, FIGO collaborated with the Nepal Society of Obstetricians and Gynecologists (NESOG) and Ministry of Health and Population to design the intervention in adherence with the national health system and training guidelines. The intervention was mainly comprised of six components: (1) Female community health volunteers and hospital staff were trained on PPFP counselling; (2) Maternity care providers were trained on PPFP counselling and PPIUD insertion and complication management; (3) Counselling aids and informational tools, leaflets, wall charts, and videos were provided and were distributed during counselling and displayed in hospital waiting areas; (4) Kelly's forceps for vaginal PPIUD insertion was provided; (5) one provider in each hospital was designated as the facility coordinator to provide on-going support for the initiative; and (6) two dedicated counsellors were recruited from July 2016 in each hospital to provide counselling [25].

All counselling services and commodities, including male condoms, pills, injectable, IUDs (Copper T 380A), and sterilization were provided free-of-charge in the essential health package provided by the Government of Nepal [26]. Though other components of the intervention

were not initially included in the essential health package, they were included following the intervention.

An evaluation study was conducted to investigate the impact and performance of the FIGO intervention in Nepal. Considering the potential benefits of the FIGO intervention to all women of reproductive ages and avoid spillover effects, the evaluation applied a cluster-randomized stepped-wedge trial design. Six local hospitals were selected as clusters that were: (1) tertiary health facilities with large catchment areas; (2) having 6,000 or more obstetric caseloads per year; and (3) not yet providing PPIUD services. Six hospitals were grouped into a pair of three based on their annual number of deliveries. Within each group, one hospital was randomized into Group 1, and the other into Group 2. The two hospitals in each group were broadly similar in terms of client socio-demographics, annual number of births, structure and human resources at the baseline. The evaluation study had an 18-month enrollment period, from September 2015 to February 2017. Group 1 hospitals and Group 2 hospitals initiated the FIGO intervention at three months and nine months, respectively, after the beginning of the enrollment data collection and maintained the intervention status since then. After the 18-month enrollment period, training of providers, monitoring and technical supervision were stopped and the number of counsellors was reduced from two to one per hospital.

## Ethics statement

This randomized trial was registered on ClinicalTrials.gov (ID number: NCT02718222) and the protocol was published [16]. The study received ethical approval from the Nepal Health Research Council (Reg no. 50/2015).

## Data collection

Women were eligible to participate in the study if they were delivering in one of the study hospitals during the 18-month enrollment period; and were residents of Nepal. Trained Nepali female enumerators approached women who gave birth before their discharge. First, enumerators gave a brief introduction of themselves and the study background and screened women for eligibility. Second, enumerators provided a consent form to eligible women and described the study purposes, participants' role, participation procedure, benefits, confidentiality, rights, and other information to make women well informed about the study. Women who consented to participation completed a written consent form and were interviewed.

75,571 out of 75,617 (99.9%) eligible women participated in the study. Enumerators interviewed participants face-to-face using a standardized questionnaire, which contained questions on women's demographic background information, the index birth and birth history, general and PPIUD specific counselling during pregnancy, and PPIUD insertion.

To evaluate the long-term impact of FIGO intervention, in July 2018, 15 months after the end of the evaluation study enrollment, the same evaluation team conducted a one-month survey at the same study hospitals, using the same questionnaire and recruiting women participants under the same inclusion and exclusion criteria. During this additional one-month survey, 4,587 out of 4,614 (99.4%) women consented to participate in the study and completed the interview (Fig 1).

## Measurements

We selected a variety of PPIUD related indicators for evaluation. Both general FP counselling and PPIUD counselling are binary variables based on women's self-reports of whether they received counselling on PPFP/birth spacing and/or PPIUD specific counselling during pregnancy or after admission to the hospital for delivery. Among women receiving PPIUD

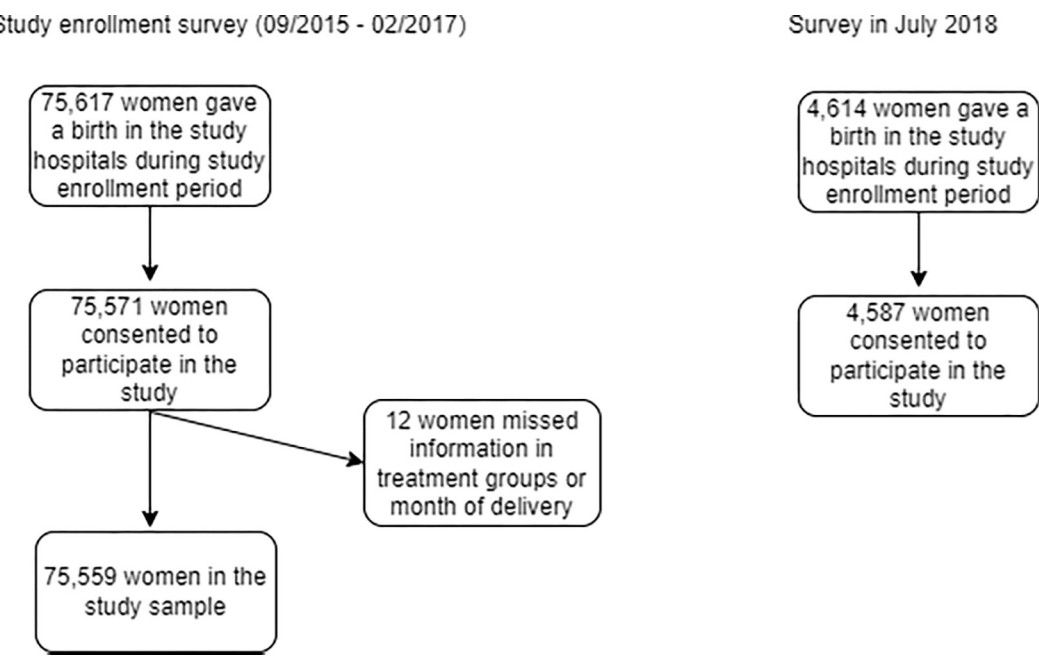

**Fig 1. CONSORT flow diagram.**

counselling, we further investigated women's timing of PPIUD counselling, PPIUD knowledge, whether women were given a leaflet during counselling, and whether women were given the opportunity to ask questions. The timing of PPIUD counselling is a categorical variable that described women receiving PPIUD counselling during ANC, after admission or both. Knowledge of PPIUD measured whether women can recall any benefits or disadvantages that were told during counselling. The questions contains multiple items of the benefits and disadvantages of using a PPIUD, such as a PPIUD being one of the most effective reversible methods to space pregnancies (one of the benefits), or the IUD may get expelled in 5% of cases (one of the disadvantages). If women can recall at least one of the benefits and at least one of the disadvantages, we categorized women as "can recall both benefit(s) and disadvantage(s)". PPIUD insertion was based on women's self-report and verified by PPIUD providers.

## Statistical methods

We analyzed the longitudinal impact of the intervention by focusing on four crucial points–the first month of the baseline (September 2015), the middle month of the intervention (July 2016 for Group 1 hospitals and October 2016 for Group 2 hospitals), the end month of the evaluation study enrollment (February 2017), and the additional one month in July 2018. We presented socio-demographic characteristics of study participants interviewed in the four crucial months and conducted Chi-square test to compare study participants in baseline with their counterparts in each of the three remaining months. We presented the distribution of all PPIUD indicators as mentioned before. By the middle month of the intervention, Group 1 and Group 2 hospitals have respectively experienced 8-month and 5-month intervention implementation. To investigate the impact of the intervention at the different implementation periods, we conducted a Chi-square test on differences in the level of PPIUD indicators across the two hospital groups in the middle month of the intervention. To investigate the long-term impact of the intervention, we used a Chi-square test on the differences in PPIUD indicators

at the end of evaluation study enrollment and in July 2018. All tests are two-tailed tests with 95% statistical significance level.

## Results

Table 1 shows the socio-demographic characteristics of the study population. In each period, there were more women aged 20–24 years than in any other age group (over 40%). Compared to study participants at baseline, study participants enrolled in the middle month of the intervention were older with a greater proportion of women aged 25 or more. This age difference is significant (p value < 0.001). Parity among women remained balanced between baseline and the middle month of the intervention with the majority of women having one child (between 55% and 60.1%) and between 31% and 34% having two children. Parity among women in the end of the enrollment period and in July 2018 differed with 60.1% and 54.1% of women respectively having one child compared to 57.9% at baseline. In each separate month, more women had secondary level education followed by above secondary level education. The

**Table 1. Selected socio-demographic characteristics of women delivering in the study hospital, by month of delivery.**

| | Baseline (09/2015) | | Middle month of the intervention during the enrollment period | | | | | End of the enrollment period | | | Jul-2018 | | |
| | | | Group 1 hospitals (07/2016) | | Group 2 hospitals (10/2016) | | | (02/2017) | | | | | |
| | % | N | % | N | % | N | P value | % | N | P value | % | N | P value |
|---|---|---|---|---|---|---|---|---|---|---|---|---|---|
| **Age group** | | | | | | | <0.001 | | | 0.164 | | | <0.001 |
| <20 | 16.1 | 780 | 12.8 | 281 | 14.3 | 364 | | 15.5 | 550 | | 11.9 | 546 | |
| 20–24 | 45.6 | 2,206 | 47.4 | 1,041 | 41.3 | 1,055 | | 44.5 | 1,577 | | 42.3 | 1,940 | |
| 25–29 | 26.1 | 1,259 | 28.4 | 623 | 30.0 | 767 | | 28.3 | 1,002 | | 30.3 | 1,391 | |
| 30 and over | 12.2 | 588 | 11.5 | 252 | 14.4 | 368 | | 11.8 | 417 | | 15.5 | 710 | |
| **Parity** | | | | | | | 0.262 | | | 0.005 | | | 0.001 |
| 1 | 57.9 | 2,796 | 55.0 | 1,208 | 57.5 | 1,468 | | 60.1 | 2,131 | | 54.1 | 2,483 | |
| 2 | 31.6 | 1,529 | 34.0 | 747 | 32.4 | 828 | | 31.4 | 1,114 | | 34.8 | 1,596 | |
| 3 and more | 10.5 | 508 | 11.0 | 242 | 10.1 | 258 | | 8.5 | 301 | | 11.1 | 508 | |
| **Women's education** | | | | | | | 0.939 | | | <0.001 | | | <0.001 |
| No schooling | 9.9 | 480 | 8.5 | 187 | 11.4 | 291 | | 6.6 | 233 | | 7.2 | 330 | |
| Primary | 10.4 | 500 | 9.4 | 207 | 10.5 | 269 | | 10.1 | 357 | | 9.9 | 455 | |
| Secondary | 45.3 | 2,191 | 46.2 | 1,016 | 45.3 | 1,157 | | 46.7 | 1,657 | | 46.5 | 2,134 | |
| Above secondary | 34.4 | 1,662 | 35.8 | 787 | 32.8 | 837 | | 36.6 | 1,299 | | 36.4 | 1,668 | |
| **Ethnicity** | | | | | | | 0.670 | | | 0.247 | | | 0.017 |
| Hill Brahmin/Chhetri | 35.6 | 1,722 | 40.3 | 886 | 33.4 | 853 | | 34.6 | 1,227 | | 35.0 | 1,603 | |
| Janajati | 38.1 | 1,840 | 30.7 | 674 | 44.6 | 1,139 | | 40.2 | 1,424 | | 36.0 | 1,653 | |
| Madhesi | 7.4 | 355 | 6.9 | 151 | 6.8 | 174 | | 6.4 | 228 | | 8.6 | 393 | |
| Dalit | 14.1 | 683 | 17.4 | 382 | 11.2 | 285 | | 14.5 | 513 | | 16.0 | 733 | |
| Muslim | 3.1 | 151 | 2.7 | 59 | 2.7 | 68 | | 2.7 | 97 | | 3.0 | 139 | |
| Others | 1.7 | 82 | 2.1 | 45 | 1.4 | 35 | | 1.6 | 57 | | 1.4 | 66 | |
| **Had Abortion(s) before** | | | | | | | 0.844 | | | 0.149 | | | 0.022 |
| Yes | 3.6 | 175 | 3.9 | 86 | 3.8 | 97 | | 3.2 | 114 | | 4.6 | 209 | |
| No | 96.4 | 4,658 | 96.1 | 2,111 | 96.2 | 2,457 | | 96.8 | 3,432 | | 95.4 | 4,378 | |
| **Male child born** | | | | | | | 0.262 | | | 0.005 | | | 0.447 |
| Yes | 54.5 | 2,636 | 52.9 | 1,163 | 54.6 | 1,394 | | 51.8 | 1,836 | | 53.8 | 2,466 | |
| No | 45.5 | 2,197 | 47.1 | 1,034 | 45.4 | 1,160 | | 48.2 | 1,710 | | 46.2 | 2,121 | |
| **Total** | **100** | **4,833** | **100** | **2,197** | **100** | **2,554** | | **100** | **3,546** | | **100** | **4,587** | |

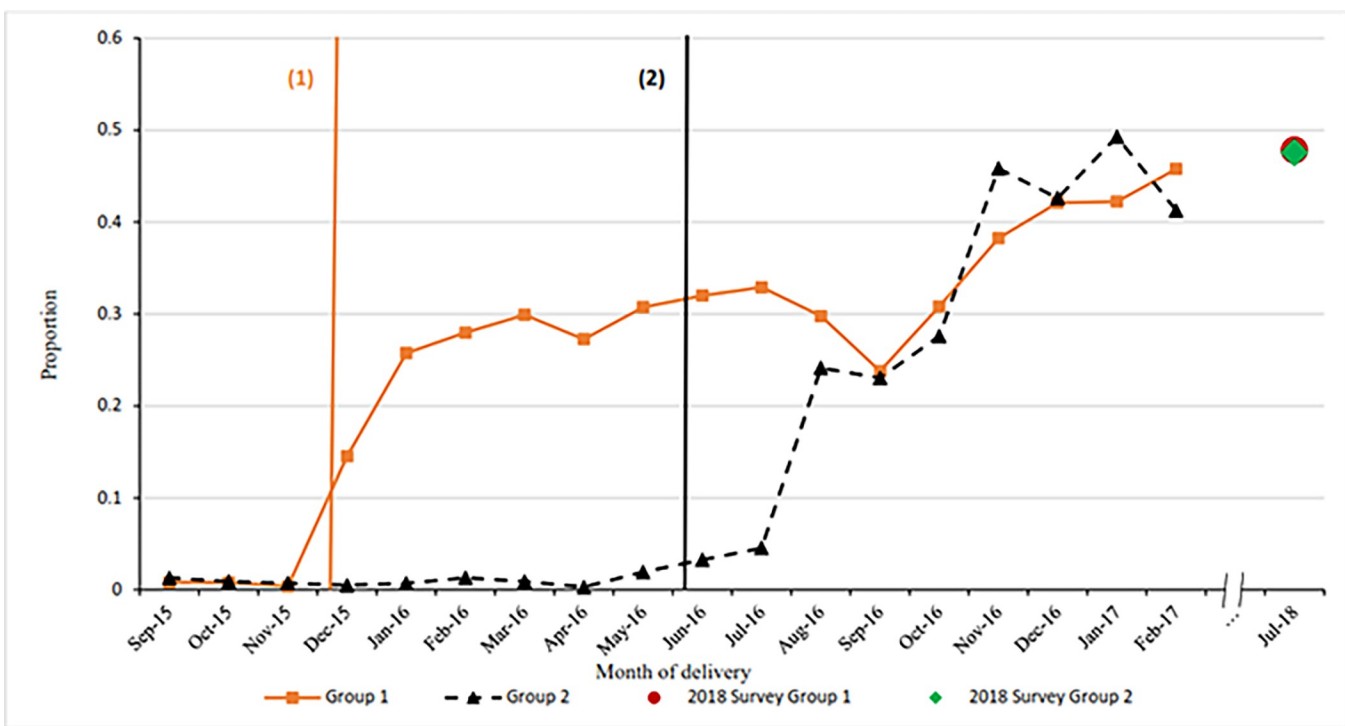

**Fig 2. Proportion of women having PPIUD counselling classified by month of delivery in Nepal.**

majority of women were either Hill Brahmin/Chhetri or Janajati. Only a small minority of women, between 3.2% and 4.6%, had previously had an abortion and over half of the women in each month had at least one male child. With the exception of parity, the background characteristics of women enrolled into the study at different months were broadly similar.

Fig 2 shows the monthly proportion of women having PPIUD counselling for group 1 and group 2 hospitals. Rates of PPIUD counselling increased after the start of the intervention and continued to increase further to the end of the enrollment period. The follow up data show the counselling rates increasing for both groups of hospitals between February 2017 and July 2018 to 47.8% for group 1 and 47.6% for group 2 from 45.8% and 41.3% respectively. This is the highest rate reached by group 1 since the start of the enrollment period; however, it is not the highest counselling rate for group 2, which was 49.3% in January 2017.

Table 2 shows the percentage of women reporting PPIUD counselling and PPIUD knowledge among those counselled during the selected months. At the baseline, 65.4% of women who reported receiving PPIUD counselling received it during ANC visits. This dropped throughout the intervention to 39.4% and 40.9% of women in group 1 and group 2 hospitals in the middle of the intervention and 39.9% of women at the end of the enrollment. At the same time the proportion of women receiving PPIUD counselling after admission increased from 19.2% at the baseline to 40.1% at the end of enrollment as did the proportion of women receiving counselling at both ANC and after admission (15.4% to 20%). In July 2018, 15 months after the end of enrollment a smaller proportion of women were receiving PPIUD counselling during ANC (20.2%), with more women receiving counselling at both instances (22.1%) or only after admission (57.7%), a statistically significant difference between February 2017 and July 2018 (P <0.001).

**Table 2. Percentage of women reporting PPIUD counselling and PPIUD knowledge, if counselled, Nepal.**

| | Baseline (09/2015) | | Middle month of the intervention during the enrollment period | | | | | End of the enrolment period (02/2017) | | July 2018 | | P value[2] |
|---|---|---|---|---|---|---|---|---|---|---|---|---|
| | | | Group 1 hospitals (07/2016) | | Group 2 hospitals (10/2016) | | P value[1] | | | | | |
| | % | N | % | N | % | N | | % | N | % | N | |
| **Timing of PPIUD counselling** | | | | | | | 0.856 | | | | | <0.001 |
| Before admission, during ANC | 65.4 | 34 | 39.4 | 285 | 40.9 | 288 | | 39.9 | 616 | 20.2 | 441 | |
| After admission only | 19.2 | 10 | 42.7 | 309 | 41.8 | 295 | | 40.1 | 618 | 57.7 | 1263 | |
| Both | 15.4 | 8 | 17.8 | 129 | 17.3 | 122 | | 20.0 | 309 | 22.1 | 484 | |
| **PPIUD knowledge** | | | | | | | <0.001 | | | | | <0.001 |
| Cannot recall any benefits/disadvantages or recall disadvantages | 32.7 | 17 | 25.6 | 185 | 39.4 | 278 | | 18.0 | 277 | 27.8 | 609 | |
| Recall any benefit(s) only | 36.5 | 19 | 55.3 | 400 | 47.8 | 337 | | 59.5 | 918 | 54.0 | 1181 | |
| Recall both benefit(s) and disadvantage(s) | 30.8 | 16 | 19.1 | 138 | 12.8 | 90 | | 22.5 | 348 | 18.2 | 398 | |
| **Women given a leaflet during counselling** | | | | | | | 0.029 | | | | | <0.001 |
| Yes | 13.5 | 7 | 57.3 | 414 | 51.5 | 363 | | 71.3 | 1100 | 55.0 | 985 | |
| No | 86.5 | 45 | 42.7 | 309 | 48.5 | 342 | | 28.7 | 443 | 45.0 | 1203 | |
| **Women given opportunity to ask questions** | | | | | | | <0.001 | | | | | <0.001 |
| Yes | 44.2 | 29 | 54.6 | 395 | 24.4 | 172 | | 59.4 | 917 | 51.2 | 1120 | |
| No | 55.8 | 23 | 45.4 | 328 | 75.6 | 533 | | 40.6 | 626 | 48.8 | 1068 | |
| **Total** | | 52 | | 723 | | 705 | | | 1543 | | 2188 | |

Note: Hospitals in Group 1 include: Koshi Zonal Hospital, Biratnagar, Morang; Lumbini Zonal Hospital; Lumbini, Rupandehi; Western Regional Hospital, Pokhara, Kaski. Hospitals in Group 2 include: Bharatpur Hospital, Bharatpur, Chitwan; Bheri Zonal Hospital, Nepalgunj, Banke; B.P Koirala Institute of Health Sciences, Dharan, Sunsari. Group 1 hospitals and Group 2 hospitals initiated the intervention respectively in December 2015 and June 2016. "End of the enrolment" refers to the end month of the enrolment data collection of women delivering in study hospitals. The International Federation of Gynecology and Obstetrics (FIGO) ended its PPIUD intervention project in June 2019 in Nepal.

[1]Chi-square test was used to compare the differences of indicator values between women interviewed in July 2016 in Group 1 hospitals and women in October 2016 in Group 2 hospitals.

[2]Chi-square test was used to compare the differences of indicator values between women interviewed in February 2017 and women in July 2018.

The baseline data in Table 2 show that 32.7% of women who were counselled could not recall any benefits or disadvantages of the PPIUD or could only report disadvantages. By the middle of the intervention 25.6% of women in group 1 and 39.4% of women in group 2 reported as such with a statistically significant difference in PPIUD knowledge between women in group 1 and group 2 hospitals (P <0.001). There was also a significant difference in PPIUD knowledge among women between the end of the enrollment and the July 2018 follow up (P <0.001). At the end of the enrollment, 18% of women in group 1 and 2 hospitals did not recall any benefits or disadvantages, or could only report disadvantages of the PPIUD, increasing to 27.8% in July 2018. Moreover, 59.5% could only recall the benefits, falling to 54% in July 2018 and 22.5% could recall both benefits and disadvantages of the PPIUD, falling to 18.2% in July 2018.

The proportion of women given a leaflet during their PPIUD counselling increased from the baseline to the middle of the intervention from 13.5% to 57.3% in group 1 and 51.5% in group 2, with a significant difference between women in group 1 and group 2 (P <0.05). This increased again at the end of the enrollment to 71.3%, though decreased significantly to 55% by July 2018 (P <0.001). Additionally, 44.2% of women at the baseline were given an opportunity to answer questions during their PPFP counselling. This increased by the middle of the intervention among women in group 1 to 54.6%, however dropped among women in group 2

**Table 3. Percentage of women receiving PPIUD counselling and PPIUD uptake, Nepal.**

| | Baseline (09/2015) | | Middle month of the intervention during the enrollment period | | | | End of the enrolment period (02/2017) | | July 2018 | | P value[2] |
|---|---|---|---|---|---|---|---|---|---|---|---|
| | | | Group 1 hospitals (07/2016) | | Group 2 hospitals (10/2016) | | P value[1] | | | | |
| | % | N | % | N | % | N | | % | N | % | N | |
| **Received any FP counselling** | | | | | | | <0.001 | | | | | <0.001 |
| Yes | 10.7 | 516 | 41.4 | 909 | 30.1 | 768 | | 45.3 | 1607 | 44.8 | 2532 | |
| No | 89.3 | 4317 | 58.6 | 1288 | 69.9 | 1786 | | 54.7 | 1939 | 55.2 | 2055 | |
| Total | | 4833 | | 2197 | | 2554 | | | 3546 | | 4587 | |
| **Received PPIUD counselling before or after admission to hospital** | | | | | | | <0.001 | | | | | <0.001 |
| Yes | 1.1 | 52 | 32.9 | 723 | 27.6 | 705 | | 43.5 | 1543 | 47.7 | 2188 | |
| No | 98.9 | 4781 | 67.1 | 1474 | 72.4 | 1849 | | 56.5 | 2003 | 52.3 | 2399 | |
| Total | | 4833 | | 2197 | | 2554 | | | 3546 | | 4587 | |
| **PPIUD uptake** | | | | | | | <0.001 | | | | | 0.05 |
| PPIUD uptake | 0.1 | 4 | 5.2 | 115 | 1.9 | 49 | | 4.3 | 151 | 3.4 | 157 | |
| Non-PPIUD woman | 99.9 | 4829 | 94.8 | 2082 | 98.1 | 2505 | | 95.7 | 3395 | 96.6 | 4430 | |
| Total | | 4833 | | 2197 | | 2554 | | | 3546 | | 4587 | |
| **PPIUD uptake among women counselled on PPIUD** | | | | | | | <0.001 | | | | | 0.002 |
| PPIUD uptake | 1.9 | 1 | 14.5 | 105 | 6.7 | 47 | | 9.6 | 148 | 6.9 | 150 | |
| Non-PPIUD uptake | 98.1 | 51 | 85.5 | 618 | 93.3 | 658 | | 90.4 | 1395 | 93.1 | 2038 | |
| Total | | 52 | | 723 | | 705 | | | 1543 | | 2188 | |

Note: Hospitals in Group 1 include: Koshi Zonal Hospital, Biratnagar, Morang; Lumbini Zonal Hospital; Lumbini, Rupandehi; Western Regional Hospital, Pokhara, Kaski. Hospitals in Group 2 include: Bharatpur Hospital, Bharatpur, Chitwan; Bheri Zonal Hospital, Nepalgunj, Banke; B.P Koirala Institute of Health Sciences, Dharan, Sunsari. Group 1 hospitals and Group 2 hospitals initiated the intervention respectively in December 2015 and June 2016. "End of the intervention" refers to the end month of the enrolment data collection of women delivering in study hospitals. The International Federation of Gynecology and Obstetrics (FIGO) ended its PPIUD intervention project in June 2019 in Nepal.

[1]Chi-square test was used to compare the differences of indicator values between women interviewed in July 2016 in Group 1 hospitals and women in October 2016 in Group 2 hospitals.

[2]Chi-square test was used to compare the differences of indicator values between women interviewed in February 2017 and women in July 2018.

to 24.4%, a statistically significant difference between the two groups (P <0.001). There was also a statistically significant difference between February 2017 and July 2018 (P <0.001), with 59.4% of women reporting they had the opportunity to ask questions in February 2017 and 51.2% in July 2018 (Table 3).

The monthly proportion of women having a PPIUD inserted is shown in Fig 3. There is evidence of PPIUD insertion rates increasing directly after the start of the intervention with month to month fluctuation and a downward trend in PPIUD insertion rates thereafter. The follow up data show a drop in PPIUD insertions from 5.3% for group 1 and 3.2% for group 2 hospitals in February 2017 to 4% and 3% July 2018. The drop in PPIUD insertions is greater among group 1 hospitals, a trend shown throughout the intervention, though the insertion rate is still higher among women in group 1 hospitals.

Fig 4 shows the proportion of women having PPIUD inserted among those counselled on PPIUD. There is very low PPIUD uptake among those counselled during the baseline period and before the start of the intervention in both groups of hospitals. Insertion rates increased immediately after the start of the intervention in group 1 hospitals, however began to increase just prior to the intervention start date among group 2 hospitals. There are variations in insertion rates among group 1 hospitals in an overall downward direction. However, PPIUD uptake

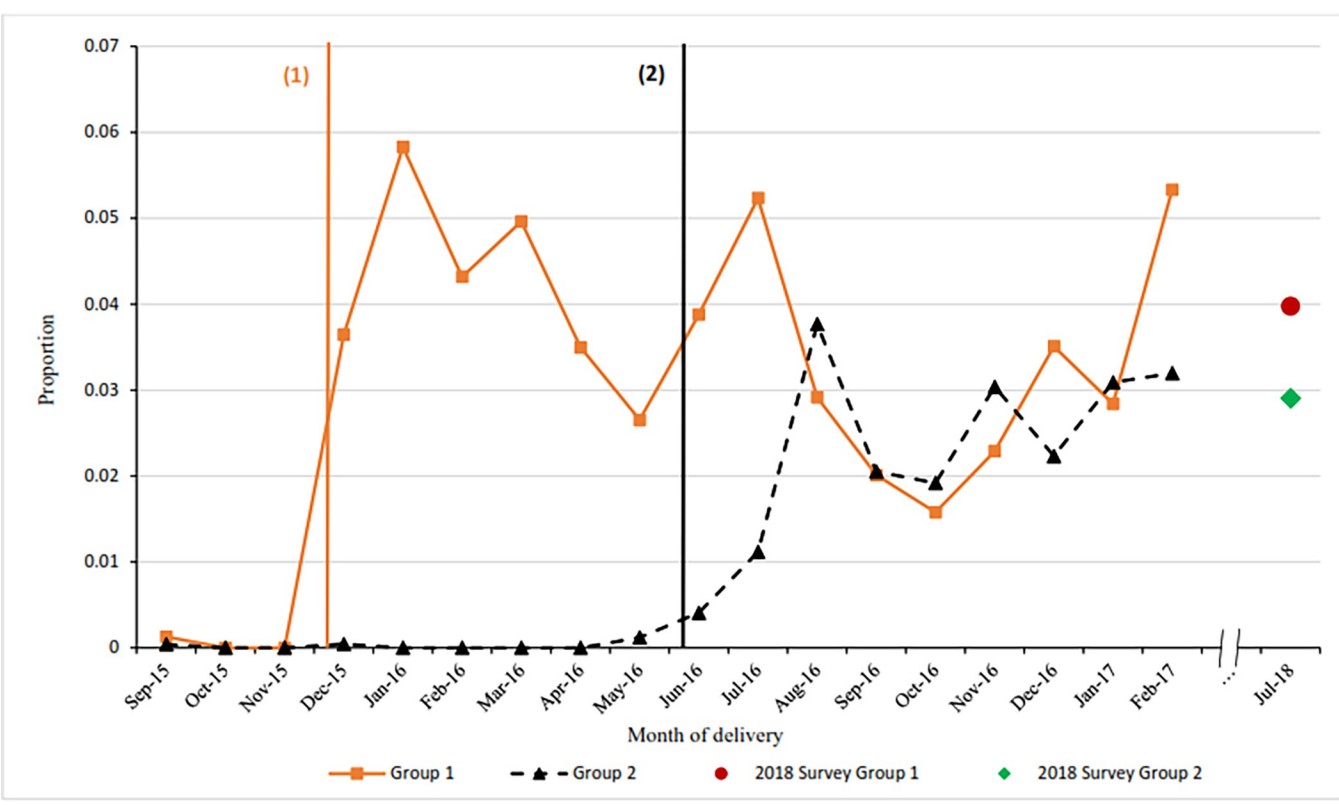

**Fig 3. Proportion of women having a PPIUD inserted classified by month of delivery in Nepal.**

in hospitals in Group 2 fell after the initial peak from the start of the intervention and continued to fall throughout the intervention, with the exception of a slight increase towards the end of the enrollment period. In keeping with the existing trend, the follow up shows a drop in PPIUD insertion rates among those counselled between February 2017 to July 2018 from 11.3% to 7.9% in group 1 and from 7.7% to 5.8% in group 2 hospitals.

Table 3 presents the percentage of women receiving PPIUD counselling and PPIUD uptake. At the baseline, 10.7% of women reported receiving any FP counselling. At the middle month of the intervention there was a statistically significant difference (P <0.001) among women reporting that they had received any FP counselling between group 1 (41.4%) and group 2 (30.1%). There was also a statistically significant difference (P <0.001) between women receiving any FP counselling between February 2017 (45.3%) and June 2018 (44.8%). Only 1.1% of women at the baseline reported receiving PPIUD counselling before or after admission to hospital. By the middle month of the PPIUD intervention this increased to 32.9% among women in group 1 and 27.6% in group 2, a statistically significant difference between the two groups (P <0.001). There was also a significant difference (P <0.001) in women receiving PPIUD counselling before or after hospital admission between February 2017 (43.5%) and July 2018 (47.7%).

PPIUD uptake among women at the baseline month was very low at 0.1%, increasing to 5.2% in group 1 and 1.9% in group 2 by the middle of the intervention, a statistically significant difference between women in group 1 and women in group 2 (P <0.001). By the end of the intervention enrollment period, 4.3% of women reported accepting a PPIUD, dropping to 3.4% in July 2018, a statistically significant drop at the 5% significance level (P ≤0.05). Among

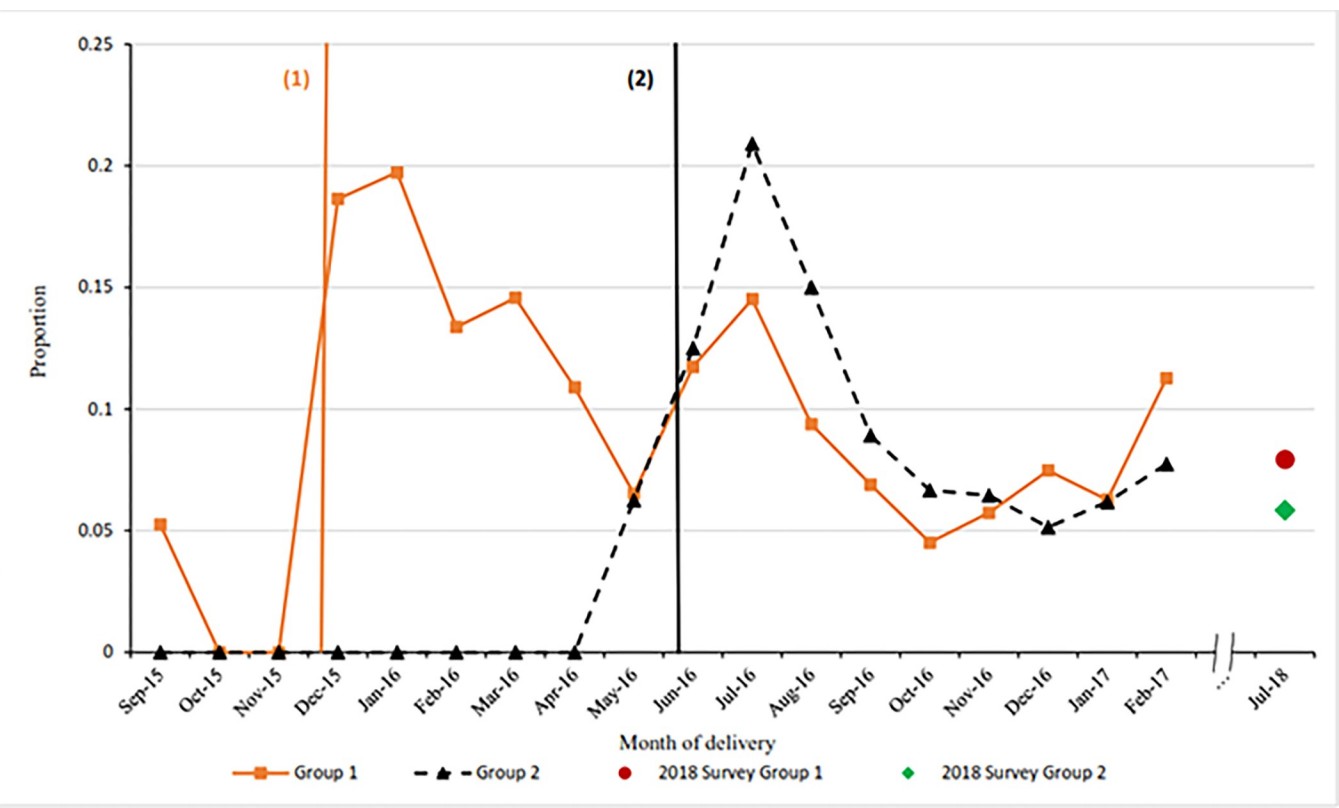

**Fig 4. Proportion of women having a PPIUD inserted among those counselled on PPIUD classified by month of delivery in Nepal.**

women who received PPIUD counselling, 1.9% reported using a PPIUD at the baseline. In group 1 this increased at the middle month of intervention to 14.5%, though only increased to 6.7% among women in group 2, a statistically significant difference in PPIUD uptake among women in the two groups of hospitals (P <0.001). The difference in PPIUD uptake among women who had previously received counselling was also statistically significant (P <0.01) between women in February 2017 and July 2018 with 9.6% of women reporting PPIUD uptake in February 2017, dropping to 6.9% in July 2018 (Table 3).

## Discussion

The results show that although the intervention had a significant impact on increasing PPIUD counselling and PPIUD insertion rates among women in the study hospitals, this impact differed between group 1 and group 2 hospitals. Furthermore, the immediate impact of the intervention is not completely long lasting with the intervention effects differing between the two groups of hospitals.

PPIUD counselling rates were fairly steady and increased throughout the intervention with a similar upward trend in counselling rates after the intervention, indicating some continued effects on counselling rates. However, by the end of the intervention enrolment period, two in five women counselled received it after admission into the hospital instead of during ANC, which increased significantly by July 2018 to four in seven women. Though a study analysing the same intervention found that women who were counselled after admission into the hospital were more likely to accept a PPIUD [15], PPFP counselling at different stages of pregnancy within the community can also increase PPIUD uptake [27,28]. As the intervention trained

providers in the hospital delivery and ANC facilities, some women, especially those not residing close to the hospital may have received their ANC elsewhere in community health settings. The increase in women receiving counselling after admission shows a need for any scale up or future intervention to train community health providers and volunteers that offer ANC, not only at the delivery hospitals. Successful links with community activities and hospitals using female community health volunteers in Nepal has been shown to actively improve PPFP counselling. However, sustainability of these linkages is paramount to continued improvements in PPFP counselling [28].

Additionally, PPIUD knowledge among women counselled changed significantly after the end of the enrolment period with a 9.8 percentage point increase in women not recalling any benefits or disadvantages of the PPIUD, or only recalling disadvantages. Poor knowledge among women on FP methods and the IUD has previously been attributed to low uptake of the method and is seen as a common barrier to accessing FP in Nepal [17,22,23]. The previous study on the intervention found that women who could not recall the benefits and disadvantages or could only recall the disadvantages of the PPIUD were less likely to accept a PPIUD [15]. Low PPIUD knowledge among women counselled during the intervention could be attributed to poor PPIUD knowledge among providers. Though PPFP knowledge among community health volunteers was found to have increased after the intervention [28] as was the case for health providers participating in this study [29]. This reduction in knowledge highlights the challenges for continued effects and, importantly, shows the need for any scale up or future intervention to implement longer term, refresher training for health workers after such interventions to ensure good quality PPFP counselling [28].

Low PPIUD knowledge among women who were counselled is also an indicator of poor quality PPFP counselling. A study examining PPIUD expulsion and discontinuation rates across six countries that implemented the intervention reported higher expulsion and discontinuation rates among women in Nepal. During a six-week follow up, Nepal had an expulsion rate of 3.9% compared to an average of 2.5% in six countries, with a discontinuation rate of 7.4% compared to an average of 3.6% in six countries [30]. The high removal rate in Nepal has been linked to several factors, including poor quality of FP counselling and misguided expectations of the device [31]. Health providers are one of the key factors that positively influence a woman's decision to choose a PPIUD when counselling has been delivered [32]. Therefore, poor quality of counselling by health providers could result in low PPIUD uptake and an increase in switching to other methods, including those that are less effective, or to abandoning the contraceptive use altogether.

PPIUD insertion among women in the study hospitals fell significantly after the intervention period. As the monthly PPIUD insertion rates for both groups of hospitals fluctuated throughout the intervention study period, the fall in July 2018 could be an instance of a downward fluctuation. However, it also hints at a lower impact of the PPIUD intervention after the enrolment period, although without further follow up data it is difficult to attribute the drop in July 2018 to a longer-term drop in insertion rates after the study period. PPIUD uptake among women who had received PPIUD counselling was higher during each study month than overall PPIUD uptake, though similarly PPIUD uptake among those counselled also dropped significantly after the intervention enrolment period. With the evidence of a downward trend in uptake among those counselled throughout the intervention, this drop could signify the withering effects of intervention beyond its intense implementation period. However, without further longitudinal data over a longer period, this cannot be fully determined.

Poor quality of counselling and continued PPIUD uptake could be linked to underlying issues in the health system in Nepal and provider's capacity to deliver care. A study examining provider perspectives on the PPIUD intervention in Nepal found that although providers were

positive about the introduction of PPFP counselling and PPIUD insertion into routine maternity care, they highlighted numerous issued with health system readiness including: 1) inadequate human resources, particularly FP counsellors, 2) lack of private space for counselling, 3) lack of IUDs, and information and counselling materials, and 4) lack of support from hospital management [33]. Health system readiness may also differ between the different hospitals within the study. The results show statistically significant variations in PPIUD knowledge, FP counselling, PPIUD counselling, PPIUD uptake and uptake among women counselled between women in group 1 and group 2 hospitals, with women in group 2 hospitals having lower PPIUD knowledge, counselling and insertion rates. This may be due to Group 2 hospitals having a shorter exposure period to the full intervention compared to Group 1, which may have affected strengthening system readiness including the providers capacity to deliver care and quality of counselling and insertion services. Thus, in order to improve PPFP counselling rates and PPIUD uptake, improvements must be made to the wider health system readiness prior to any interventions and longer-term exposure to the intervention. Furthermore, more research should be conducted to examine the key factors contributing to the significant differences in counselling and insertions rates at hospital level.

The study has some limitations. Only tertiary level hospitals with high obstetric caseloads were included ($> 6,000$ cases a year) in the study, excluding women who had delivered outside the formal healthcare system and delivered in smaller primary health care centers or at home. It is uncertain if the intervention, as implemented, would be suited to lower-level district hospitals, primary health centers or other health posts that are used by women in more rural areas. Access to health facilities is difficult for many women, especially those in more rural and remote areas, therefore many women may be reluctant to accept PPIUD services as side-effects and removal would require a visit to a health facility [16]. Another limitation is that this study looks at counselling and insertion and not counselling and acceptance or consent to the PPIUD insertion. This limits the analysis to those who had the PPIUD inserted and does not account for those who had consented to having a PPIUD but were unable to procure it, due to the unavailability of a trained provider and/or the PPIUD at the time of delivery, for example. The latter would be a better indicator to monitor PPIUD counselling impact in terms of generating the demand for PPIUD. The women in the study were also not nationally representative. On average, they were younger and had more years of schooling than women of reproductive age in Nepal [12]. With regards to assessing the longer-term impact of the intervention, the sample using data from July 2018 has given a glimpse of a single selected calendar month that was 15 months after the end of study enrolment and a rigorous implementation of the intervention. However, a longer observational period is needed to determine the sustained impact of the intervention beyond its implementation period. Due to the lack of data, we were unable to consider information critical to explaining the levels and trends over time and by hospital group. More specifically, no information was available on any stock out of supplies, including PPIUD, availability of trained staff for IUD insertion at the time of delivery, transfer of trained staff from study to non-study hospitals and on other potential confounders. Lastly, the follow up study in July 2018 did not ascertain providers' perspectives nor their capacity to continue to provide counselling and PPIUD insertions. Therefore, we were unable to determine if the changes were due to weakening interest among women in PPIUD or because of slackening performance of providers.

The previous study of the intervention validates the feasibility of the approach used to integrate PPFP counselling and PPIUD insertion into routine maternity care services in Nepal and shows a demand for PPIUD among women who desire to space or limit their births [15]. This study complements the findings by examining the differences in PPFP counselling and PPIUD insertion between the two study hospital groups and by investigating the levels 15 months

after the study enrolment period and implementation. The intervention had an appreciable impact on PPIUD counselling rates throughout the study period and during the follow up, and had an impact on PPIUD uptake, especially among those who had received PPIUD counselling throughout the study period, though this impact was less notable in July 2018 follow up. Additionally, the impact the intervention had on counselling and uptake was dissimilar between group 1 and group 2 hospitals. The intervention had an encouraging impact in the study hospitals with a greater impact in group 1 hospitals. Information and counselling on PPFP and PPIUD is key to informed choice and uptake of PPFP and PPIUD. Despite an increase in the counselling rate, the levels are nowhere close to universal. While PPFP uptake is an individual choice, the provision of information and counselling is clearly within the remit of policies and programmes that must be fully implemented. The efficiency of the intervention may be improved by expanding the coverage of PPFP counselling, providing refresher PPFP counselling training to providers and ensuring that counselling is of a higher standard. The study also raises some questions about the continued impact of the intervention that need to be explored further with the longitudinal data over several months beyond the implementation period to assess the long-term impact of the intervention.

## Conclusions

The PPIUD intervention in Nepal increased PPIUD counselling and insertion rates, especially insertions among those who had received PPIUD counselling. However, PPIUD counselling and insertion rates varied significantly between women in group 1 and group 2 hospitals, with women in group 2 having lower PPIUD counselling and insertion rates, possibly due to the shorter period of exposure to intense implementation of the intervention. PPIUD counselling rates were shown to increase after the study period, though at the same time our results suggest that PPIUD insertions rates were falling, therefore indicating that the intervention effects were not completely maintained after the implementation ended. Placing our findings in the context of Nepal and with those from other studies, our study suggests that the longer term impact of the intervention could potentially be enhanced by 1) expanding the coverage of PPFP counselling to include health facilities that provide pregnancy care in the community, and 2) providing refresher training for providers, especially on PPFP counselling. In addition, the intervention period should be longer with the aim of improving the effects that were evident from group 1 hospitals that had a longer period of intervention than group 2. These recommendations will likely improve health service readiness and increase the capacity of providers to deliver good quality and sustainable counselling and insertion services. In order to determine the long-term effectiveness and sustainability of the intervention beyond the implementation, a longer period of follow up of study hospitals is required.

## Supporting information

**S1 File. CONSORT checklist.**
(DOC)

**S2 File. Study protocol.**
(PDF)

## Acknowledgments

We gratefully acknowledge the valuable support provided by the PPIUD NESOG's Nepal team and FIGO team in London during the data collection of study. Our thanks also go to the study participants for sharing their personal information, opinions and experiences.

## Author Contributions

**Conceptualization:** Mahesh Chandra Puri, Lucy Stone, Iqbal H. Shah.

**Data curation:** Mahesh Chandra Puri, Muqi Guo, Lucy Stone.

**Formal analysis:** Muqi Guo.

**Funding acquisition:** Iqbal H. Shah.

**Methodology:** Mahesh Chandra Puri, Muqi Guo, Iqbal H. Shah.

**Project administration:** Mahesh Chandra Puri.

**Resources:** Iqbal H. Shah.

**Supervision:** Mahesh Chandra Puri, Iqbal H. Shah.

**Validation:** Muqi Guo, Lucy Stone, Iqbal H. Shah.

**Visualization:** Lucy Stone.

**Writing – original draft:** Mahesh Chandra Puri.

**Writing – review & editing:** Mahesh Chandra Puri, Muqi Guo, Lucy Stone, Iqbal H. Shah.

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
