## [Decision Letter · Decision Letter 0]

21 Jul 2022

PGPH-D-22-00849

Integrating postpartum IUD counselling and insertion into routine maternity care in Nepal: Are the effects sustainable?

Dear Dr. Puri,

Thank you for submitting your manuscript to PLOS Global Public Health. After careful consideration, we feel that it has merit but does not fully meet PLOS Global Public Health’s publication criteria as it currently stands. Therefore, we invite you to submit a revised version of the manuscript that addresses the points raised during the review process.

Please note that we have only been able to secure a single reviewer to assess your manuscript. We are issuing a decision on your manuscript at this point to prevent further delays in the evaluation of your manuscript. Please be aware that the editor who handles your revised manuscript might find it necessary to invite additional reviewers to assess this work once the revised manuscript is submitted. However, we will aim to proceed on the basis of this single review if possible. 

We look forward to receiving your revised manuscript.

Kind regards,

Julia Robinson

Executive Editor

Journal Requirements:

1. Please provide a detailed online Financial Disclosure statement. This is published with the article. It must therefore be completed in full sentences and contain the exact wording you wish to be published.

a. Please clarify all sources of funding (financial or material support) for your study. List the grants (with grant number) or organizations (with url) that supported your study, including funding received from your institution. 

b. State the initials, alongside each funding source, of each author to receive each grant.

c. State what role the funders took in the study. If the funders had no role in your study, please state: “The funders had no role in study design, data collection and analysis, decision to publish, or preparation of the manuscript.”

d. If any authors received a salary from any of your funders, please state which authors and which funders.

2. Please update your online Competing Interests statement. If you have no competing interests to declare, please state: “The authors have declared that no competing interests exist.”

3. Please provide separate figure file in .tif or .eps format and ensure that all files are under our size limit of 10MB.

4. Please ensure that you refer to Figures 1 and 3 in your text as, if accepted, production will need these references to link the reader to the figures.

5. We have noticed that you have uploaded Supporting Information files, but you have not included a list of legends. Please add a full list of legends for your Supporting Information files after the references list.

Additional Editor Comments (if provided):

Reviewer #1: Integrating PPIUD counselling and insertion into routine maternity care in nepal: are effects sustainable?

Overall

Thank you for your manuscript. The data is valuable and some of the interpretations are interesting but there is some clear information missing which needs to be addressed before conclusions can be made. Please see my detailed comments.

As I understand it there are two sets of analysis - one is looking at comparing the two groups and the second is looking at changes over time. As I right in thinking there was no change in the intervention and things were being implemented in the same way in 2017 as they were in July 2018? If this is the case, then the discussions about sustainability are not really valid, as you cant discuss sustainability if the support was not removed. As the latter would be looking at what happens once all interventions are removed. Maybe it would be better to centre the discussion around a longitudinal analysis of the intervention over time and remove the word and discussion about sustainability.

Please highlight as a limitation the fact that you looked at counselling and insertion and not counselling and acceptance/consent. The latter is a more direct result of counselling where as insertion has a further rate-limiting factor which you are not accounting for, which is the availability of a trained provider to insert at the time of delivery. In other words amongst this data you cannot ascertain the number of women consenting who did not end up receiving PPIUD even though they may have wanted one.

Abstract

Please amend the abstract according to the suggested changes in the detailed review of the manuscript as I don't believe some of your conclusions hold.

Eg: 'The results suggest that the impact and sustained effects of the intervention could be enhanced by regular follow up and supervision, expanding the coverage of PPFP counselling to include other health facilities, and improving the training for providers, especially on providing high quality PPFP counselling.'

The results don't suggest that. You have no data to suggest that the sustainability would be improved by regular follow up and supervision etc… This is your opinion or recommendation. You need to be clear about what your results show and what you suggest/advise.

Background

Line 105 - What is the purpose of comparing the two groups? Why does it matter. Please expand. What are the differences between the two groups? Are there any structural differences? It seems to me that the main difference is the implementation period. Please highlight this.

Line 106 - What do you mean by '15 months after the end of the evaluation study enrolment'. What is the significance of this time period. Was that the end of the intervention period? What interventions ended at this point? Did funding end? Did training of staff end at this point? What support was withdrawn? Please explain in more detail. It seems to me that later on you mention the intervention is still ongoing until June 2019.

Data and Methods

Line 116

Please reference FIGO's PPIUD initiative rather than just explaining it in your own words without a reference. This paper should help: 'Planning and implementation of a FIGO postpartum intrauterine device initiative in six countries' Linda de Caestecker et al in the IJGO

Line 132 The evaluation applied a cluster-randomized stepped-wedge trial design.

Why did you choose this design? Why were the hospitals grouped as such. What was the significance betweenthe two groups. Please give more information on the details of each of the hospitals in the two groups. Were they comparable in terms of structure? Human Resources / number births / demographics of women using the hospitals / facilities for counselling and insertion etc…

Line 139 You mention the intervention ended in June 2019. What was the purpose of analysing the effect after 'the enrolment period'. What changed after the end of this period? Did the interventions change in some way? If not, then I suggest your analysis is more of an interpretation over time, looking at the longitudinal impact of the intervention over all the months.

Line 124

I don't believe the FIGO PPIUD initiative supplied Cu IUDs - these were supplied by the government only and not through the initiative. Please use information re implementation from the implementers descriptive paper. This will ensure it is accurate.

Statistical methods

Line 190 - 193: Please change this paragraph and describe it as a longitudinal assessment at 4 points. The way it is currently described makes it look like the final assessment is at the end of the intervention, which I think is not the case?

Results

Line 2123: Regarding demographics, I suggest you run a statistical analysis comparing your analysis groups to your baseline group as otherwise it is not certain that the groups are similar enough for comparison. Parity as you say is grossly dissimilar in July 2018. Why did this happen? You have interviewed a cohort of women with very different contraceptive intentions than you have at baseline. It makes comparing them impossible. One possibility would be to separate out from your other groups the Para 1s and seeing if your differences hold? If you cant do this it may be better to leave this months data out completely.

Line 272 - I think you mean figure 3 not 2.

Tables 2 and 3

Unless there is something specific that happened in 02/2017 Im not sure of the significance of comparing that point to one month in July 2018? Stats looking at longitudinal changes may be more valuable?

Discussion

Line 345 - You may be interested in three further studies looking exactly at the issue you mention. The more recent studies involved one of your group 1 hospitals (Koshi) and reported on the involvement of Female Community Health Volunteers, which improved uptake and outcomes in counselling and uptake in the surrounding hospitals. Please read these studies as they are relevant to your points about ANC counselling vs in hospital counselling.

K Thapa, R Dhital, S Rajbhandari, S Acharya, S Mishra, S Mani Pokhrel, S Pande, E-A Tunnacliffe, A Makins. Factors affecting the behavior outcomes on post-partum intrauterine contraceptive device uptake and continuation in Nepal: a qualitative study. BMC Pregnancy and Childbirth 2019; 19: 148

Rolina Dhital, Ram Chandra Siwal, Khem Narayan Pokhrel, Sabina Pokhrel, Heera Tuladhar, Suzanna Bright, Emily-Anne Tunnacliffe, Kusum Thapa, Anita Makins. Evaluating the impact of female community health volunteer involvement in a postpartum family planning intervention in Nepal: a mixed-methods study at one-year post-intervention. PLOS ONE. Published October 20, 2021. Available: https://doi.org/10.1371/journal.pone.0258834

K. Thapa, R. Dhital, Sameena Rajbhandari, Sangeeta Mishra, Shanti Subedi, Bhogendra Dotel, Sapana Vaidya, Saroja Pande, Emily-Anne Tunnacliffe, Anita Makins and Sabaratnam Arulkumaran. Improving post-partum family planning services provided by female community health volunteers in Nepal: a mixed methods study. BMC Pregnancy and Childbirth 2020; 20:123

Line 368 - I think you mean PPIUD insertion among women in study hospitals dropped as there is no information on consent. Consenting and receiving PPIUD are two different things. My understanding is that you measured insertion. Please correct this.

Line 368 - Your one month follow up group is entirely different to your other groups as 99% are Para 1s - i.e. only just had one baby. In low income countries the outcome of that birth is more uncertain and much fewer women who have only just given birth to their first child are likely to accept contraception. You are not comparing similar groups of women.

Paragraph 368 to 384 - I don't think you can suggest a true difference between PPIUD insertion rates in the group based on two major flaws:

1. The fact that there is such a high level of fluctuation from month to month and you only have a snap shot of one month.

2. The fact that the groups are different. Your 2018 group is 99% Para 1s. These women are not comparable to your original group.

You mention these as limitations but in actual fact in makes it impossible to draw any firm conclusions. As mentioned earlier you could try and make a comparison between Para 1s in the other groups in order to get round issue 2. Issue 1 remains. IT be worth just removing that data?

Line 396

The data doesn't suggest this. You are suggesting that as a possible explanation - that the group 2 hospitals may have had these issues. Please word this correctly. And please expand and substantiate your suggestion with information regarding the differences in the structures of the hospitals in the two groups. What was the staff to patient ratio? Which hospitals had individual private counselling spaces? Which hospitals had lack of IEC material or shortages of IUDs? What were their management structures? If you don't have this information please explain this as a limitation of your work and hence ability to interpret the results.

Line 412 to 415

I would go as far as to say that the data from July 2018 is not assessing sustainability as my understanding is that intervention was still on going. The question is why its impact was not as great as first off.

Line 418

What is the purpose of comparing the two groups? My understanding is that it was a cluster randomized study as so the two groups would not be different. If this is the case what is the value in comparing the two groups? Is it the length of time of implementation from your analysis? If so, please say this.

Line 420

The study as it stands is not sufficient to look at sustainability. Please amend.

Line 424/5

'…impact was notable and seemingly deteriorated during the follow up'. You are not able to conclude this with the methodology as it stands.

Line 429 There is little information in your study about the training of providers and how it was lacking. I suggest you expand on this in the methodology section. As its difficult to suggest changes when you haven't included any information on what the training involved and how it could be of a higher standard. Perhaps you are referring to results from another study? If so please explain this.

Conclusion

Line 440 - 445 Same issue as I have raised above.

Line 445 - From the information provided in the paper, this is the only plausible explanation for the differences in the 2 groups and should be highlighted as the main conclusion of that analysis. The remainder you mention is your opinion, and needs to be explained as such. Unless you are able to substantiate the other claims with more data / evidence.

Figures

Fig 3 and 4 - the variations from one month to the next are enormous in both groups. There must be some implementation reason for this as the counselling rates are much more smooth - eg were their sufficient trained inserters at all times day and night (that may explain the drops) / were their stock outs of IUDs (would also explain the drops), were insertions immediately postpartum(ie in the labour room) or within 48hrs meaning availability of an area on the ward to do it may have limited insertion? Please explain further. If you do not have this information please explain as a limitation when interpreting these two graphs.
---

## [Decision Letter · Decision Letter 1]

13 Jan 2023

PGPH-D-22-00849R1

Integrating postpartum IUD counselling and insertion into routine maternity care in Nepal:  Assessing trends over time

Dear Dr. Puri,

Thank you for submitting your manuscript to PLOS Global Public Health. After careful consideration, we feel that it has merit but does not fully meet PLOS Global Public Health’s publication criteria as it currently stands. Therefore, we invite you to submit a revised version of the manuscript that addresses the points raised during the review process.

We look forward to receiving your revised manuscript.

Kind regards,

Dickson Abanimi Amugsi, PhD

Academic Editor

Journal Requirements:

Additional Editor Comments (if provided):

Reviewers' comments:

Reviewer's Responses to Questions

**Comments to the Author**

1. If the authors have adequately addressed your comments raised in a previous round of review and you feel that this manuscript is now acceptable for publication, you may indicate that here to bypass the “Comments to the Author” section, enter your conflict of interest statement in the “Confidential to Editor” section, and submit your "Accept" recommendation.

Reviewer #2: All comments have been addressed

Reviewer #3: (No Response)

Reviewer #4: All comments have been addressed

2. Does this manuscript meet PLOS Global Public Health’s publication criteria? Is the manuscript technically sound, and do the data support the conclusions? The manuscript must describe methodologically and ethically rigorous research with conclusions that are appropriately drawn based on the data presented.

Reviewer #2: Yes

Reviewer #3: Partly

Reviewer #4: Yes

3. Has the statistical analysis been performed appropriately and rigorously?

Reviewer #2: Yes

Reviewer #3: Yes

Reviewer #4: Yes

4. Have the authors made all data underlying the findings in their manuscript fully available (please refer to the Data Availability Statement at the start of the manuscript PDF file)?

Reviewer #2: Yes

Reviewer #3: Yes

Reviewer #4: Yes

5. Is the manuscript presented in an intelligible fashion and written in standard English?

Reviewer #2: Yes

Reviewer #3: No

Reviewer #4: Yes

6. Review Comments to the Author

Reviewer #2: This is an important topic both in that uptake of PPIUD is needed, and also that it is important for us to look at if, and also what can make, our interventions sustainable without program support. This article increases the knowledge in some areas. Notable to me is that if teaching maintains and uptake decreases something must have changed in the teaching...

Reviewer #3: This manuscript describes an analysis looking at long-term impact of an intervention to promote PPIUD in Nepal. The manuscript has been edited in response to reviewer comments once; I have some additional suggestions below:

Abstract

See point below.

Introduction

It would seem that the framing of this manuscript could relate more directly to the aims of the FIGO intervention FIGO intervention to increase the capacity of health care professionals to offer PPIUDs by training community midwives, health workers, doctors and delivery unit staff, as appropriate. In other words, the purpose of the 15 month follow up was to see how well the providers were maintaining their capacity to officer PPIUDs. In this instance, a mixed methods study would be very helpful to understand not only what provider performance outcomes are but also why they have changed.

Methods

Line 141: specially leaflets??? Is this a typo?

Line 148-152: What other "contraceptives" are we talking about? This is focused only on IUDs, correct? This paragraph is repetitive and needs to be edited to remove duplication.

Line 169-174: This paragraph can be simplified as follows:

After the 18-month

170 enrollment period, training of providers, monitoring and technical

173 supervision were stopped and the number of counsellors was reduced from two to one per

174 hospital.

Line 174-177: delete this section since it is repeated in paragraph immediately after

Results

Many of the results related to baseline and intervention have already been reported. See Reprod Health. 2019;16(1), 69 (main study paper); it seems some of the figures in this manuscript are duplicative of what is already published; see specifically Figure 3 PPIUD counseling and Figure 5, PPIUD uptake. Please explain how the figures in the two papers are different.

It seems this analysis is most useful by focusing on the 15 month followup survey to the main intervention. I suggest revising the manuscript to reflect that point and remove all redundancies that are already reported in the main study paper.

Discussion

Importantly, seeing a drop-off of the intervention once implementation has ended is very common. The more interesting question is why has this happened so framing the discussion more concisely and coherently around this topic would make it easier for the reader to understand.

Line 383-384: What is "it"? Seems to refer to different studies; confusing since talking about study results in first part of sentence. Though PPFP knowledge among

383 health workers is shown to increase after the intervention, it has also been shown to reduce as

384 time goes by [28, 29].

Line 400: PPIUD insertion among women in the study hospitals fell significantly after the intervention

401 enrolment.

I think just saying the "intervention period" rather than "enrollment" will make it easier for readers to understand.

Conclusion

I agree with reviewer 1 that the issue of comparability of the two groups is important particularly since party and education are significantly different from the intervention group.

The conclusion that the difference in hospital group uptake related to the length of intervention implementation is purely conjecture since no other attributes to hospital service delivery were assessed.

Reviewer #4: The article was well written and worthy of publication however, there is need for gramatical review and editing.

For example:

line 222- change "in baseline" to "at baseline"

line 230 - 231: the statement "More proportion...to the baseline" seems incomplete.

7. PLOS authors have the option to publish the peer review history of their article (what does this mean?). If published, this will include your full peer review and any attached files.

**Do you want your identity to be public for this peer review?** For information about this choice, including consent withdrawal, please see our Privacy Policy.

Reviewer #2: **Yes: **Rondi E Anderson

Reviewer #3: No

Reviewer #4: No

---

## [Editor Report · Decision Letter 2]

8 Feb 2023

Integrating postpartum IUD counselling and insertion into routine maternity care in Nepal:  Assessing trends over time

PGPH-D-22-00849R2

Dear Dr. Puri,

We are pleased to inform you that your manuscript 'Integrating postpartum IUD counselling and insertion into routine maternity care in Nepal:  Assessing trends over time' has been provisionally accepted for publication in PLOS Global Public Health.

Best regards,

Dickson Abanimi Amugsi, PhD

Academic Editor

Thank you for adequately addressing the reviewer's comments. The manuscript is a well written piece of work. My main concern was the reviewer's observation that some aspects of the work were a duplication of an earlier published paper. This can be a serious issue, leading to rejection of a manuscript if the authors fail to justify why it is the case. You have been able to explain this very clearly, hence my decision to accept your paper for publication.

I just wanted to point out that your statistical analysis appears a bit basic (at least, confidence intervals should have been included), though it addressed your research question to a very large extent. Looking at the sample size, one would have expected a more advance statistical modelling. You may consider undertaking a rigorous statistical analysis in the future.